# Plocabulin, a Novel Tubulin Inhibitor, Has Potent Antitumour Activity in Patient-Derived Xenograft Models of Soft Tissue Sarcoma

**DOI:** 10.3390/ijms23137454

**Published:** 2022-07-05

**Authors:** Yannick Wang, Agnieszka Wozniak, Jasmien Cornillie, Pablo Avilés, Maria Debiec-Rychter, Raf Sciot, Patrick Schöffski

**Affiliations:** 1Laboratory of Experimental Oncology, Department of Oncology, KU Leuven, 3000 Leuven, Belgium; agnieszka.wozniak@kuleuven.be (A.W.); jasmien.cornillie@zol.be (J.C.); patrick.schoffski@kuleuven.be (P.S.); 2PharmaMar S.A., 28770 Madrid, Spain; paviles@pharmamar.com; 3Department of Human Genetics, University Hospitals Leuven, KU Leuven, 3000 Leuven, Belgium; maria.debiec-rychter@kuleuven.be; 4Department of Pathology, University Hospitals Leuven, KU Leuven, 3000 Leuven, Belgium; raf.sciot@uzleuven.be; 5Department of General Medical Oncology, University Hospitals Leuven, 3000 Leuven, Belgium

**Keywords:** soft tissue sarcoma, patient-derived xenograft, plocabulin, PM060184

## Abstract

A clinically relevant subset of patients with soft tissue sarcoma presents with either locally advanced or upfront metastatic disease, or will develop distant metastases over time, despite successful treatment of their primary tumour. The currently available systemic agents to treat such advanced cases only provide modest disease control and are not active in all histological subtypes. Thus, there is an unmet need for novel and more efficacious agents to improve the outcome of this rare disease. In the current preclinical in vivo study, we evaluated plocabulin, a novel tubulin inhibitor, in five distinct histological subtypes of soft tissue sarcoma: dedifferentiated liposarcoma, leiomyosarcoma, undifferentiated sarcoma, intimal sarcoma and *CIC*-rearranged sarcoma. The efficacy was tested in seven patient-derived xenograft models, which were generated by the engraftment of tumour fragments from patients directly into nude mice. The treatment lasted 22 days, and the efficacy of the drug was assessed and compared to the doxorubicin and vehicle groups by volumetric analysis, histopathology and immunohistochemistry. We observed tumour volume control in all the tested histological subtypes. Additionally, in three sarcoma subtypes, extensive central necrosis, associated with significant tumour regression, was seen. This histological response is explained by the drug’s vascular-disruptive properties, reflected by a decreased total vascular area in the xenografts. Our results demonstrate the in vivo efficacy of plocabulin in the preclinical models of soft tissue sarcoma and corroborate the findings of our previous study, which demonstrated similar vascular-disruptive effects in gastrointestinal stromal tumours—another subtype of soft tissue sarcoma. Our data provide a convincing rationale for further clinical exploration of plocabulin in soft tissue sarcomas.

## 1. Introduction

Soft tissue sarcomas (STS) are a highly heterogeneous group of solid tumours of mesenchymal origin and represent around 1% of all adult malignancies [1]. Over 100 distinct histological subtypes have been described by the World Health Organisation [2]. The treatment of STS is primarily based on the extent of the disease at presentation: localised tumours are commonly treated surgically (±radiotherapy) with curative intent, while patients with locally advanced and/or metastatic disease typically receive systemic, often palliative, therapy. At diagnosis, 10–15% of patients have a locally advanced or metastatic disease [3,4]. For those with resectable local disease, around a quarter of all patients who undergo successful treatment for their primary tumour eventually develop metastasis, despite the initial curative intent. This fraction increases to around half of the patients with high-grade tumours [5]. Those with metastatic disease face a dire prognosis with a median overall survival of 12–19 months [6].

The first line of therapy for advanced disease has been doxorubicin for over 40 years, despite providing a modest response rate of around 15%. Multiple clinical trials have since been carried out to evaluate either doxorubicin combined with other cytotoxic agents or regimens with other chemotherapeutic compounds to improve clinical outcomes, though none of them were able to show a reproducible benefit in overall survival [7,8,9,10]. Thus, the therapeutic repertoire for STS remains rather limited. Beyond first-line therapies, systemic treatments are typically administered with a palliative intent to delay further disease progression. Although several agents are available in this setting, they only provide low response rates and disease control in the range of a few months [11,12,13,14]. Notably, several of these cytotoxic agents (e.g., eribulin, docetaxel, trabectedin) are tubulin-binding agents and target microtubules, which are a key constituent of the cytoskeleton [15].

Plocabulin (PM060184, PharmaMar, Madrid, Spain) is a novel tubulin-binding agent, originally isolated from the marine sponge *Lithoplocamia lithistoides* [16]. It showed potent antitumour activity, both in the in vitro and in vivo models of pancreas, breast, colorectal, gastric, prostate and kidney carcinoma [17,18,19]. It interacts with the maytansine binding site on β-tubulin, which leads to microtubule disorganisation and fragmentation in pulmonary adenocarcinoma tumour cells, and in prometaphase cell arrest and apoptosis [17]. Apart from the antitubulin activity of the compound, preclinical work by Galmarini and colleagues demonstrated potent antiangiogenic properties, reflected by a strong reduction in vascular volume, both in vitro in the cultures and spheroids of human umbilical vein endothelial cells, and in vivo in a xenograft of lung adenocarcinoma [20]. Our recent in vivo study confirmed the strong vascular-disruptive potent antitumour effects of the drug in the patient-derived xenograft (PDX) models of gastrointestinal stromal tumours, a distinct, common histological subtype of STS [21]. Moreover, a phase I study demonstrated promising antitumour effects of plocabulin in patients with advanced colorectal adenocarcinoma, gastrointestinal stromal tumour and cervix, breast, thymus, lung and parotid adenocystic carcinoma [22].

In the current study, we aimed to expand the histological panel of sarcoma types treated with the experimental agent and test its efficacy in seven well-characterised and self-made additional PDX models, of which the model characteristics and donor patient characteristics have been reported earlier [23]. We present here, the in vivo efficacy of plocabulin in five additional histological subtypes of STS, including both the common and rare variants comprising dedifferentiated liposarcoma, leiomyosarcoma, undifferentiated sarcoma, intimal sarcoma and *CIC*-rearranged sarcoma.

## 2. Results

### 2.1. All PDX Models Were Histologically Stable

All seven PDX models (comprising dedifferentiated liposarcoma, leiomyosarcoma, undifferentiated sarcoma, intimal sarcoma and *CIC*-rearranged sarcoma), established using tumour tissue from patients with STS and which had gone through mouse passaging for 4 to 20 passages, showed stable histological characteristics as compared to the original donor tumours. The model features and donor characteristics were described previously [23]. A comparison of the original patient tumour and passages used in the current in vivo experiment can be seen in Figure 1. All models retained their main morphological features and, if applicable, their immunohistochemical characteristics, which is a typical finding of the xenograft platform used for the work described here.

### 2.2. No Severe Toxicity Was Associated with Plocabulin Treatment

Three out of forty-one plocabulin-treated mice were lost during the treatment period: one had to be sacrificed due to tumour burden, and two were found dead without prior symptoms. Four mice were lost in the vehicle groups: three were sacrificed due to a high tumour burden, and one was sacrificed due to illness (a loss of body weight and a curved back). The doxorubicin administered i.p. at 3 mg/kg body weight was associated with severe toxicity: 23 out of the 41 doxorubicin-treated mice were lost before day 22. Thirteen were found dead, four were sacrificed due to illness, and six were sacrificed due to tumour dimensions. A detailed overview of the body weight evolution and the number of mice is shown in Appendix A. Ultimately, 49 tumours were lost before day 22 (7 in the vehicle groups, 36 in the doxorubicin groups, and 6 in the plocabulin groups) and, therefore, excluded from further analyses.

### 2.3. Plocabulin Resulted in Tumour Shrinkage in LMS, CIC and USTS Models, and in Tumour Volume Stabilisation in DDLPS and IS

The plocabulin treatment resulted in a statistically significant tumour volume reduction to 27%, 83%, 40% and 90% of the baseline volume in UZLX-STS22_2F^LMS^, -STS111^LMS^, -STS134^CIC^ and -STS84X^USTS^, respectively. The tumour volume stabilisation, after 22 days, was observed in the other models. Tumours of -STS112^DDLPS^, -STS124^DDLPS^ and -STS122F^IS^ reached 110%, 140% and 130% of the starting volumes, respectively, changes which were statistically not significant. Of note, the standard of care agent, doxorubicin, did not result in any effect on the tumour volume in any of the models. A comparison of relative tumour volumes at the end of the experiment showed statistically significant differences between the vehicle and plocabulin groups in all seven models, further supporting an antitumour effect. Figure 2 and Table 1 show a detailed overview of the tumour volumes and statistical analyses.

### 2.4. Plocabulin Treatment Induced Necrosis and Degenerative Changes in All Histological Subtypes

The plocabulin-treated tumours showed enhanced necrosis and degenerative changes in all histological subtypes. The most extensive changes were seen in the LMS, CIC and USTS models, with pronounced central necrosis and degeneration. Despite these extensive regressive changes, a remaining, thin rim of viable tissue was observed at the periphery of the tumours, which are characteristic changes for tumours treated with vascular disrupting agents. In this residual, albeit small rim, the proliferative activity was comparable to the vehicle-treated tumours. Regressive changes were also seen in the DDLPS and IS models, albeit to a lesser extent, with only focal necrosis and degenerative changes. An overview is given in Figure 3.

Due to this extensive histological response, only a limited number of plocabulin-treated tumours could be evaluated, as the histological analysis was solely carried out in non-necrotic areas: UZLX-STS22_2F^LMS^: none out of 10 tumours; UZLX-STS111^LMS^: 2 out of 10; UZLX-STS84X^USTS^: none out of 10; UZLX-STS122F^IS^: 8 out of 13; UZLX-STS134^CIC^: none out of 7. All plocabulin-treated DDLPS tumours could be analysed due to the less-pronounced necrotic changes. In the evaluable models, plocabulin decreased the total vascular areas in UZLX-STS111LMS and -STS122FIS and decreased the mean vascular density in UZLX-STS111^LMS^. Conversely, no antivascular effects were observed in the DDLPS models. A detailed overview of the results of the histological analysis can be found in Figure 4 and Figure 5. 

## 3. Discussion

We conducted a preclinical study of plocabulin in seven PDX models of STS, evaluating its antitumour efficacy in five different histological subtypes of mesenchymal tumours.

The antitumour activity of the experimental agent was observed in all models and all histological subtypes. In the LMS, USTS and CIC models, the plocabulin treatment resulted in tumour shrinkage, while in the DDLPS and IS models, tumour volume stabilisation was seen. A direct histological correlate was observed in the form of enhanced tumour necrosis and tissue degeneration in all models, with the most extensive changes in the LMS, USTS and CIC models.

In the latter models, these regressive changes were so pronounced that this rendered further histological analyses impossible in the vast majority of the *ex*-mice tumours. A few tumours, however, contained sufficient residual, non-necrotic tissue, which allowed further analyses. A decrease in the total vascular area was observed, along with a decrease in the mean vascular density, suggesting a vascular-disruptive mechanism for the response. Interestingly, even in the most necrotic and degenerative tumours, a small though viable rim of the tumour remained at the periphery of the lesions, surrounding a large necrotic core. The same finding was observed in our previous study with plocabulin in the PDX of gastrointestinal stromal tumours [21]. This pattern of regression is typically seen in solid tumours treated with vascular disrupting agents [24]. While the central part of the tumours becomes necrotic as the vasculature is remodelled, the outer rim of the tumours may still obtain oxygen and critical nutrients from nearby vessels located in the surrounding normal tissue, or through local diffusion [25]. This observation raises the notion that additional treatment modalities may be indicated to target this residual viable rim, as, over time, it may regrow and lead to a loss of disease control. Further in vivo studies and, possibly, combinations of this vascular-disruptive agent with other compounds with activity in STS are warranted to explore this hypothesis.

Thus, based on our experiments, it seems that the response to the plocabulin treatment is determined by the histological subtype, with the most promising results in LMS, CIC and USTS. This is in line with the increasing notion that STS are a heterogeneous group of malignancies, not only histologically but also pertaining to the response to systemic treatment. Multiple histological subtypes have been pooled together in historical clinical trials [10,11,26,27] due to the rare nature of STS, facilitating the initiation of such investigations. However, as each histological subtype was represented by a limited number of patients, such studies were underpowered to reliably perform subgroup analyses according to the histological subtype. Consequently, it remained challenging to adequately determine against which histological subtypes the assessed systemic treatment was truly efficacious. It should be noted that tumour volume control, as observed in the DDLPS and IS models, is also a meaningful endpoint aside from tumour regression, as observed in the other models. PFS with the stabilisation of disease is one of the most commonly used major endpoints in clinical trials involving advanced STS, as objective responses in later lines of treatment are exceedingly rare [28,29].

To our knowledge, no studies exploring the in vivo mechanisms of resistance to plocabulin have been published yet. However, Pantazopoulou and colleagues investigated the molecular basis of resistance to plocabulin in vitro, using a fungal model in Aspergillus nidulans, taking into account that “the roles of MTs [microtubules] in *A. nidulans* resemble those in mammalian cells” [30]. They reported that the mutations in eukaryotic translation initiation factor 2B (eIF2B) conferred resistance to plocabulin in vitro. Furthermore, eIF2B plays a central role in the integrated stress response (IRS), a pathway in eukaryotic cells that allows cells to restore homeostasis following various stress stimuli [31]. It is highly conserved between fungi and mammalian cells, and a gain-of-function mutation in its alpha subunit leads to a hyperactivation of the IRS pathway, which results in resistance to plocabulin [30]. Interestingly, EIF2B1, the gene coding for eIF2Bα, is located on chromosome 12q24, a region that is commonly overrepresented in DDLPS [32]. Future studies are needed to unravel the mechanisms of resistance and reveal the potential biomarkers that can predict sensitivity to plocabulin, which will aid in selecting patients who will likely benefit from the plocabulin treatment.

Despite being the current first-line standard of care in settings of advanced STS, we could not show any meaningful impact on the tumour volume of doxorubicin in the seven different sarcoma models in the current study. This finding is concordant with the very modest efficacy of doxorubicin observed in clinical settings [9]. Interestingly, the donor patients of models UZLX-STS124^DDLPS^ and -STS22_2F^LMS^ received doxorubicin after a donation of tumour tissue for the establishment of the PDX models. Both patients progressed under the systemic therapy [23], which is recapitulated by our PDX models, illustrating once again the predictive value of patient-derived xenografts in preclinical drug testing.

In the current study, we observed a high rate of death in the groups treated with doxorubicin, which was administered i.p. A total of 23 mice were lost out of a total of 68 mice in this group (death rate of 34%). Gilliams et al. have previously demonstrated that doxorubicin i.p. causes oxidative stress and lipid oxidation in the peritoneal cavity, resulting in phrenic weakness and drug-induced peritonitis. Conversely, these adverse effects were not seen if the drug was administered i.v. [33]. Additionally, the extravasation of doxorubicin during i.v. injection results in the necrosis of subcutaneous tissue [34]. Thus, it seems plausible that similar effects could occur in the abdominal cavity and that doxorubicin should be administered i.v. instead, to reduce toxicity.

A facet that could be considered is the replacement of the human tumour stroma with a murine one after successive passaging. Repeated implantation through passaging and engraftment replaces the native human tumour stroma with a murine one [35] and, thus, also the vessels, which may affect the efficacy of treatments. Previous studies have, however, demonstrated that plocabulin efficiently targets vessels of both human and murine origin [20,21].

A limitation of the current study is the collection of the tumours only at the very end of the experiment—the setup we had successfully used in our previous in vivo studies when studying STS in the PDX models [36,37,38]. This approach did not pose a challenge for the histological evaluation, as we had never observed such striking necrotic tissue changes as in the current study. In the current study, however, many plocabulin-treated tumours had to be excluded from the histological analysis since HPF fully containing viable tumour were required for microscopic evaluation. A potential solution could be the collection of percutaneous biopsies throughout the experiment, or the sacrification of some mice within experimental groups, earlier during the experiment. This method would support the aforementioned suggestion of combining plocabulin with other cytotoxic or targeted agents, as this could, in theory, lead to total necrosis of the tumours, rendering a histological analysis impossible using the current method.

## 4. Materials and Methods

### 4.1. Animals

Six-week-old female, athymic Rj:NMRI-Foxn1*^nu/nu^* mice were purchased from Janvier Labs. The animals were housed in groups of 4 in individually ventilated cages. Autoclaved wood chips (Lignocel^®^ BK 8/15, J. Rettenmaier & Soehne GmbH, Rosenberg, Germany) were used as bedding material, and cotton cylinders (Cocoon^®^, Dates and Group, Riga, Latvia) were provided as nesting material. A commercial diet in pellet form (ssniff^®^ R/M-H, sniff Spezialdiäten GmbH, Soest, Germany) and autoclaved tap water were given ad libitum. The mice were housed under 12-h light/dark cycles with a room temperature of 22 °C (±2 °C).

### 4.2. Xenograft Models

The in vivo efficacy of plocabulin was evaluated using 7 STS PDX models, characterised previously [23]. These represent 5 different histological subtypes of STS: dedifferentiated liposarcoma (DDLPS; UZLX-STS112^DDLPS^ and -STS124^DDLPS^), leiomyosarcoma (LMS; UZL-STS22F^LMS^ and -STS111^LMS^), undifferentiated soft tissue sarcoma (USTS; UZLX- STS84X^USTS^), intimal sarcoma (IS; UZLX- STS122F^IS^) and *CIC*-rearranged sarcoma (CIC; UZLX-STS134^CIC^). These xenografts were established, as described previously, by the bilateral implantation of tumour fragments from donor patients directly into the subcutaneous tissue of nude mice [39]. Once the tumour growth was observed and the tumour sizes reached ethical limits, the tumour-bearing mice were sacrificed, the tumours were collected, and fragments of the tumours were engrafted into a new generation of mice (‘passaging’). The models were considered as ‘established’ when the stable histological and molecular features were confirmed in at least two passages after the tumour engraftment from the donor patients into mice. The models had gone through different mouse passages before being used for the current experiment. Passages 13, 11, 20, 12, 4, 11 and 10 of UZLX-STS112^DDLPS^, -STS124^DDLPS^, -STS22F^LMS^, -STS111^LMS^, -STS84X^USTS^, -STS122F^IS^ and -STS134^CIC^ were used, respectively. The collection of human tumour tissue and its usage for xenografting was approved by the Medical Ethics Committee of the University Hospitals Leuven (approval number: S53483). All donor patients gave written informed consent for the use of their tissue for this purpose. The Animal Ethics Committee of KU Leuven approved the creation of the PDX models and the in vivo experiments (approval number: P175–2015), which were performed according to the local guidelines and Belgian regulations.

### 4.3. Drug Preparation

Plocabulin and its non-active vehicle were obtained as lyophilized vials from PharmaMar S.A. The working solutions were prepared under sterile conditions, according to the manufacturer’s instructions. Doxorubicin hydrochloride was purchased from Sigma-Aldrich (D1515). The working solutions were prepared with sterile water for injection to reach a concentration of 0.5 mg/mL. The aliquots were kept at −80 °C, and individual vials were thawed for each dosing. All solutions were brought to room temperature before administration.

### 4.4. Experimental Design

For the cohort expansion, a total of 140 mice were implanted bilaterally and subcutaneously with tumour fragments that were generated after passaging under general anaesthesia. Ultimately, 211 tumours were generated, as not all implanted tumour fragments resulted in tumour growth. Each mouse was engrafted with a tumour from a single model. The tumour-bearing mice were randomised into 3 treatment groups per model: (i) the vehicle given intravenously (i.v.), once weekly (QW); (ii) doxorubicin 3 mg/kg given intraperitoneally (i.p.), QW; and (iii) plocabulin 16 mg/kg i.v., QW. Both the vehicle and plocabulin were administered by bolus injection into the lateral tail veins. The tumours were measured three times per week using a digital calliper during the 22-day treatment period. The body weights were measured daily, and the animals’ well-being was checked daily. Mice were sacrificed for ethical reasons: in case of severe body weight loss (>20% of starting body weight) or if the tumour burden exceeded 2000 mm^3^. On the final day of drug administration, all mice were euthanised by cervical dislocation after an intraperitoneal injection of pentobarbital sodium (Vetoquinol, Fort Worth, TX, USA). The tumours were then collected, partly snap-frozen in liquid nitrogen and partly fixed in 10% neutral buffered formalin for further analyses.

### 4.5. Tumour Volume Measurement

The tumours were approximated as ellipsoids, and volumes were calculated using the following formula: volume = 4/3 × π × length/2 × width/2 × height/2, with a length defined as the greatest dimension of the tumours and the other two axes perpendicular to the previous ones. To take into account the different starting volumes of the tumours, the volumes were standardised as relative tumour volumes: relative tumour volume = (volume on day x)/(volume on day 1) × 100%. The tumours from the mice sacrificed before the final day of the experiment were excluded from the volumetric analysis.

### 4.6. Immunohistochemistry

Immunohistochemical staining for the mouse double minute 2 homolog (MDM2) was used to demonstrate the MDM2 overexpression in the DDLPS models as the oncogenic driver and to confirm the stability of the models. Caldesmon and α-smooth muscle actin were used as smooth muscle markers in the LMS models. CD99 and Wilms’ tumour protein (WT1) expression were used to confirm the stability of the *CIC*-rearranged sarcoma model. Ki-67 and phospho-histone H3 (pHH3) were used as markers of cell proliferation. Cleaved poly-ADP-ribose-polymerase (cleaved PARP) was used as an immunohistochemical marker of apoptosis. Sections of 4 µm were cut for the haematoxylin-eosin (H&E) and immunohistochemical staining from the formalin-fixed and paraffin-embedded tumours. Tissue sections were deparaffinized and rehydrated in UltraClear^TM^ (Klinipath, Olen, Belgium) and ethanol series, respectively. The endogenous peroxidase activity was either quenched by a 15-min treatment with 0.9% hydrogen peroxidase dissolved in distilled water for CD 31, or a 20-min treatment with 0.09% hydrogen peroxidase in methanol for all the other antigens. The epitopes were unmasked with a heat-induced epitope retrieval using a citrate buffer (pH = 6.0) for Ki-67, MDM2 and pHH3. A Tris-EDTA buffer was used for αSMA, caldesmon, CD31, CD99 and WT1. Reveal Decloaker (Biocare Medical, Pacheco, CA, USA) was used for cleaved PARP. The following primary antibodies were used for the immunohistochemistry: Ki-67 (clone SP6, MA5-14520, Thermo Fisher Scientific, Ready To Use, Waltham, MA, USA), MDM2 (clone IF2, 337100, Life Technologies, 1:50), pHH3 (9701, Cell Signaling Technologies, 1:100, Danvers, MA, USA), αSMA (clone 1A4, M085101, DAKO, 1:500), caldesmon (clone E89, ab32330, Abcam, 1:100, Cambridge, UK), CD31 (clone SZ31, DIA-310, Dianova, 1:100, Hamburg, Germany), CD99 (clone SP119, ab227738, Abcam, 1:200), WT1 (clone SP321, ab224806, Abcam, 1:100) and cleaved PARP (clone 5E1, ab32064, Abcam, 1:100). EnVision^®^+ System-HRP Labelled Polymer Anti-Rabbit (K4003, DAKO, Ready To Use) was used as a secondary antibody for Ki-67, pHH3, caldesmon, CD99 and WT1. EnVision^®^+ System-HRP Labelled Polymer Anti-Mouse (K4001, DAKO, Ready To Use) was used as a secondary antibody for MDM2 and αSMA. Biotinylated Goat Anti-Rat IgG Antibody, mouse adsorbed (BA-9401, Vector Laboratories, 1:100, Newark, CA, USA), was used as a secondary antibody for CD31. SignalStain^®^ Boost IHC Detection Reagent (8114, Cell Signaling, Ready To Use) was used as a secondary antibody for cPARP. The antigen-antibody complexes were visualised using diaminobenzidine (Dako, Glostrup, Denmark), and the slides were counterstained with Gill’s haematoxylin (VWR, Radnor, PA, USA).

### 4.7. Histological Assessment

The mitotic and apoptotic activities of the treated tumours were evaluated by counting the number of mitotic figures and apoptotic cells in 10 high-power fields (HPF, 400-fold magnification, 0.105 mm^2^ area) on H&E. The Ki-67 index was calculated as the average percentage of the Ki-67 positive cells in 5 images captured at 400× magnification. Immunohistochemistry for pHH3 and cleaved PARP were used as additional markers for proliferation and apoptosis, respectively, and were assessed by calculating the average number of positive cells in 10 HPF. CD31 stains were used to assess tumour vasculature. The mean vascular density and the total vascular area were defined as the average number of vessels and the average area of those vessels covered on 5 digital micrographs captured at 200× magnification (0.418 mm^2^ area), respectively. Individual microvessels were identified as described previously [40]. The percentage of tumour necrosis/degeneration was calculated as follows: (area of necrosis/degeneration) ÷ (total tumour area) × 100%. All slides were scanned using the Philips IntelliSite Ultra Fast Scanner, and the images were obtained and analysed on the Philips IntelliSite Pathology Solution platform. Only tumours collected from mice sacrificed on day 22 of drug exposure were included in the histological assessment.

### 4.8. Statistical Analysis

The absolute tumour volumes on day 1 vs. day 22 of the treatment were compared using the Wilcoxon matched-pairs signed-rank test to categorise the tumour volume evolution into stabilisation, shrinkage or growth. Tumour shrinkage or growth was defined as a statistically significant difference, whereas stabilisation was defined as the cases where no statistically significant difference could be detected using the aforementioned test. A comparison of the non-parametric variables of the different treatment groups (relative tumour volumes at the end of the experiment and histological analyses) was done using a Kruskal–Wallis test with Dunn’s multiple comparisons test post hoc. In the case of only two evaluable groups, the Mann–Whitney U test was used instead. Two tumours engrafted onto the same mouse were regarded as not dependent statistical events and were analysed as separate lesions [23]. GraphPad Prism 9 (GraphPad Software, San Diego, CA, USA) was used for all analyses, with a *p* < 0.05 considered statistically significant. The confidence intervals were calculated using the Student’s t distribution in Microsoft Excel 2019 (Microsoft, Redmond, WA, USA).

## 5. Conclusions

In conclusion, we demonstrated the efficacy of the novel tubulin inhibitor plocabulin in seven PDX models of other STS, spanning five histological subtypes. We confirmed its efficacy, corroborated the findings of our previous in vivo study, and provided a convincing preclinical rationale for further clinical exploration of the agent.

## Figures and Tables

**Figure 1 ijms-23-07454-f001:**
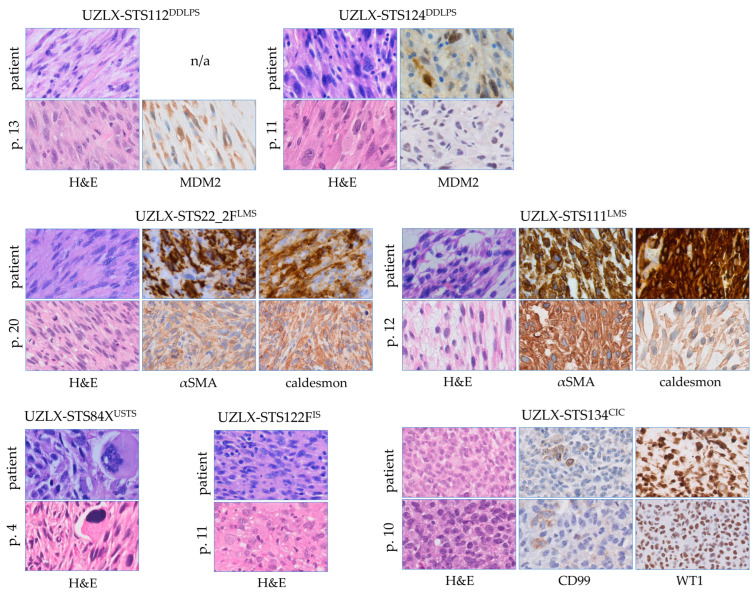
All 7 patient-derived xenograft models retained the histological features of the original tumours and, if applicable, their characteristic immunohistochemical profiles. Comparison of *ex*-mice tumours and the original patients’ tumours. Haematoxylin-eosin and immunohistochemical stains were captured at 200× magnification. CIC: *CIC*-rearranged sarcoma; DDLPS: dedifferentiated liposarcoma; H&E: haematoxylin-eosin; IS: intimal sarcoma; LMS: leiomyosarcoma; n/a: not available; USTS: undifferentiated sarcoma.

**Figure 2 ijms-23-07454-f002:**
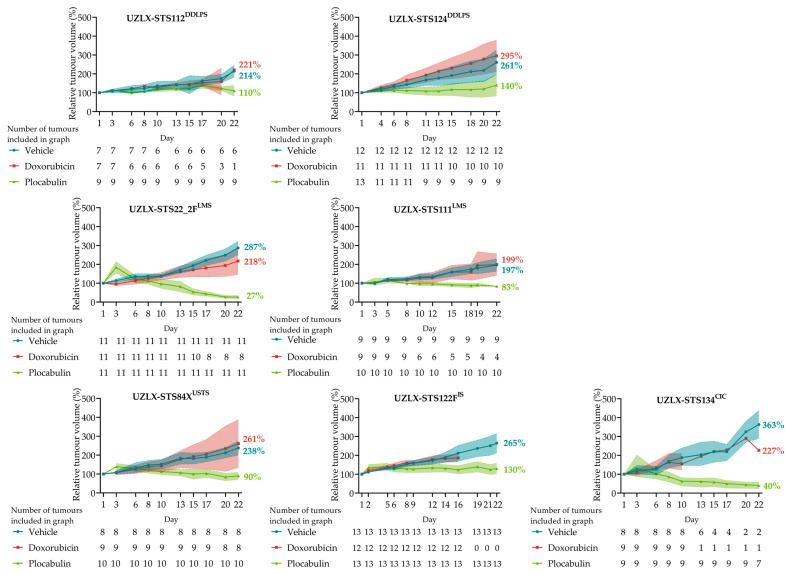
Plocabulin treatment resulted in tumour volume control in 5 histological subtypes. Tumour volume stabilisation was observed in the DDLPS and IS models, while tumour regression was observed in the LMS, CIC and USTS models. Tumour volume evolution, shown as mean with 95% confidence interval (shaded) (except for UZLX-STS134^CIC^: mean with range), with the number of tumours included on each respective day below each individual graph. CIC: *CIC*-rearranged sarcoma; DDLPS: dedifferentiated liposarcoma; IS: intimal sarcoma; LMS: leiomyosarcoma; USTS: undifferentiated sarcoma.

**Figure 3 ijms-23-07454-f003:**
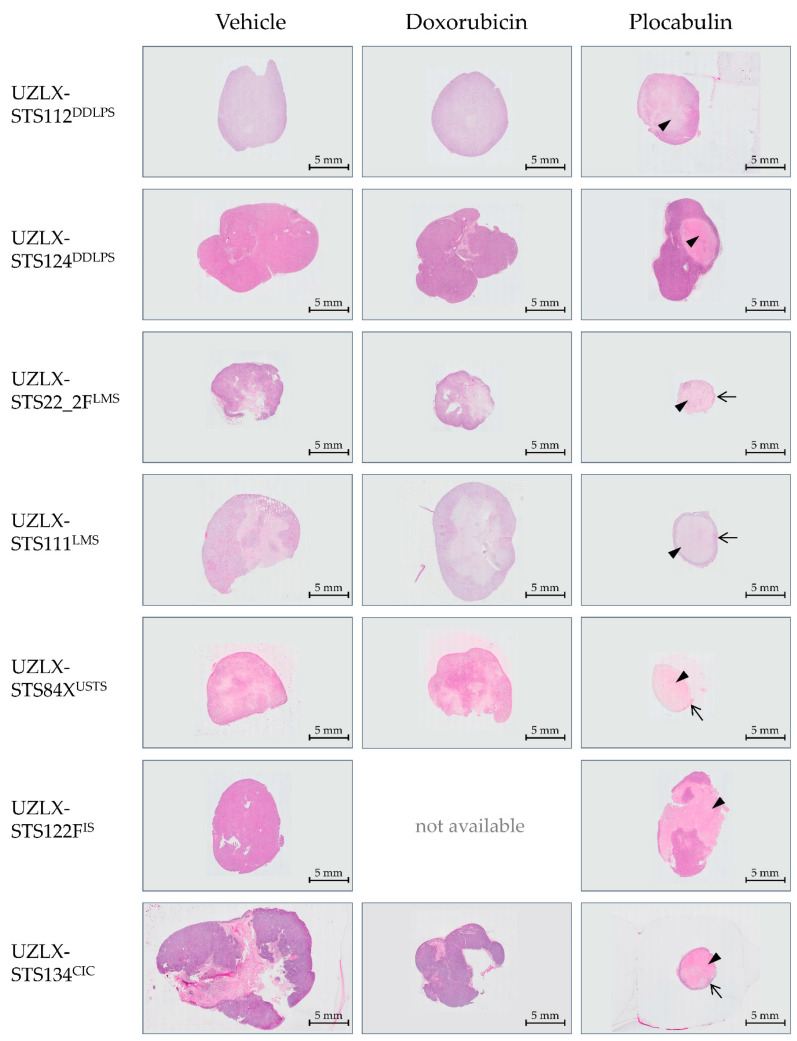
Tumours treated with plocabulin showed pronounced necrosis and regression (arrowheads), predominantly in the centre of the lesions. The most profound changes were seen in the LMS, CRS and USTS models, in which only a small rim of viable residual tumour remained at the periphery (arrows). No doxorubicin-treated mice of UZLX-STS122F^IS^ were available as all mice were found dead due to the strong toxicity of the drug when administered intraperitoneally. Representative haematoxylin and eosin stains (2.5× magnification) of tumours after 22 days of treatment. CIC: *CIC*-rearranged sarcoma; DDLPS: dedifferentiated liposarcoma; IS: intimal sarcoma; LMS: leiomyosarcoma; USTS: undifferentiated sarcoma. Micrographs were taken at 2.5× magnification.

**Figure 4 ijms-23-07454-f004:**
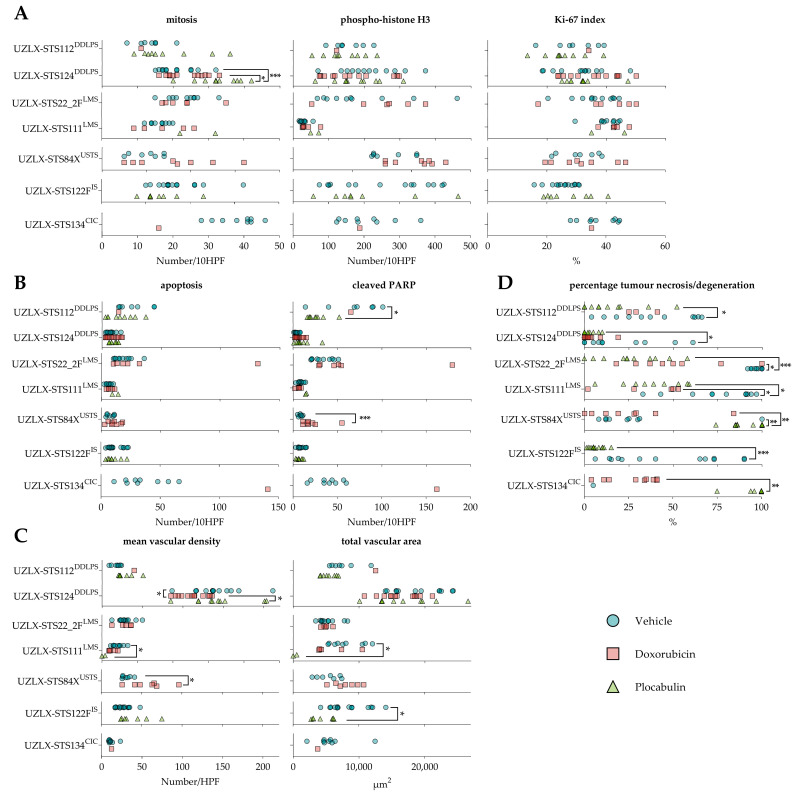
Plocabulin showed strong vascular-disruptive activity, resulting in a decreased total vascular area in UZLX-STS111^LMS^ and UZLX-STS122F^IS^. Additionally, a decreased mean vascular density was observed in UZLX-STS111^LMS^. Due to the extensive necrosis, only a fraction of the plocabulin-treated tumours could be evaluated histologically: UZLX-STS22_2F^LMS^: none out of 10; UZLX-STS111^LMS^: 2 out of 10; UZLX-STS84X^USTS^: none out of 10; UZLX-STS122F^IS^: 8 out of 13; UZLX-STS134^CIC^: none out of 7. None of the doxorubicin-treated tumours of UZLX-STS122F^IS^ could be evaluated since all tumour-bearing mice were lost before the end of the experiment due to the toxicity of the drug. (**A**) Proliferative activity. More mitotic figures were observed in the plocabulin-treated tumours of UZLX-STS124^DDLPS^ than in the vehicle-treated and doxorubicin-treated ones, although no correlation could be demonstrated with the phospho-histone H3 stain or Ki-67 index. (**B**) Apoptotic activity. Plocabulin-treated tumours of UZLX-STS112^DDLPS^ showed less cleaved PARP positive cells than vehicle-treated ones, although no correlation could be found with the apoptotic count or the other DDLPS model. (**C**) Angiogenesis. Tumour vasculature was visualised with immunohistochemistry using a CD31-antibody, and the number of vessels and their luminal areas was measured. Plocabulin-treated tumours of UZLX-STS111^LMS^ showed a lower mean vascular density. (**D**) Percentage of tumour necrosis/degeneration. Plocabulin-treated tumours of all histological subtypes showed enhanced tumour necrosis/degeneration compared to vehicle. Data shown as individual values per tumour. All statistically significant differences are annotated. The *p*-values displayed were either obtained with the Mann–Whitney U test (in case of only 2 evaluable groups) or Kruskal–Wallis’ test with Dunn’s multiple comparisons test *post hoc* (in case all 3 groups were evaluable). *: *p* < 0.05; **: *p* < 0.01; ***: *p* < 0.001; CIC: *CIC*-rearranged sarcoma; DDLPS: dedifferentiated liposarcoma; IS: intimal sarcoma; LMS: leiomyosarcoma; PARP: Poly (ADP-ribose) polymerase; USTS: undifferentiated sarcoma.

**Figure 5 ijms-23-07454-f005:**
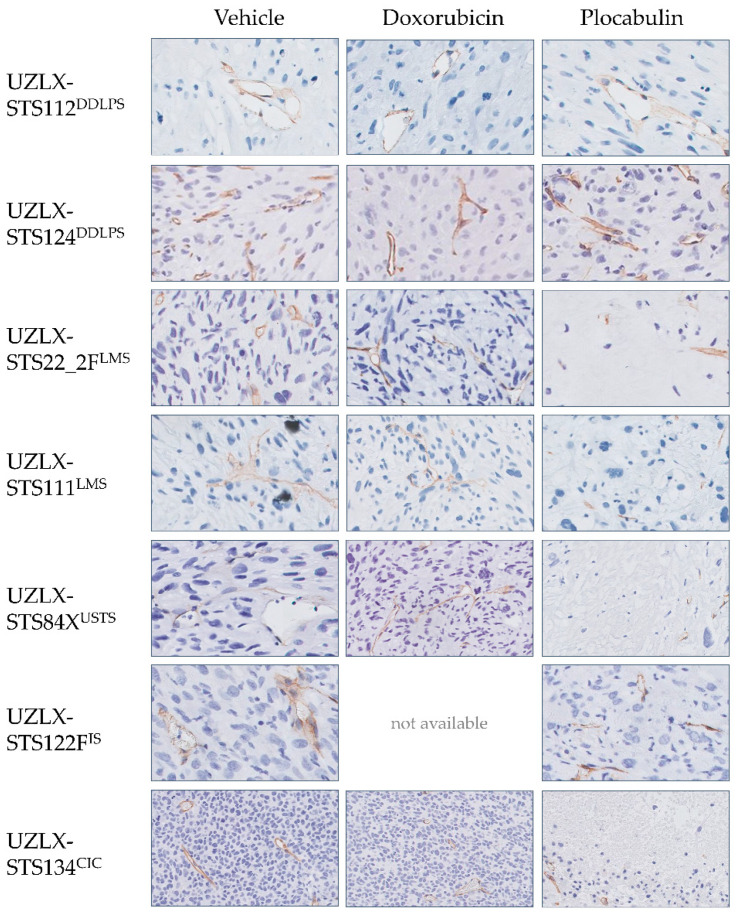
Plocabulin showed vascular-disruptive effects. Plocabulin-treated tumours contained more small and collapsed vessels lumina, while vehicle-treated tumours mostly contained large and ectatic blood vessels, often filled with erythrocytes. No images of doxorubicin-treated tumours are available because all doxorubicin-treated mice were found dead on day 19, and thus all tumours in that group were lost. Representative CD31 stains captured at 200× magnification. CIC: *CIC*-rearranged sarcoma; DDLPS: dedifferentiated liposarcoma; IS: intimal sarcoma; LMS: leiomyosarcoma; USTS: undifferentiated sarcoma.

**Table 1 ijms-23-07454-t001:** Plocabulin treatment resulted in statistically significant tumour regression in the LMS, CRS and USTS models and in tumour volume stabilisation in the DDLPS and IS models. Detailed overview of the average absolute tumour volumes and average relative tumour volumes per group.

		Group	Average Absolute Tumour Volume			Average Relative Tumour Volume			
			Day 1, mm^3^	Day 22, mm^3^	*p*-Value Compared to Day 1 (WMP)	Day 22 (95% CI), %	*p*-Value of Kruskal–Wallis Test	*p*-Value Compared to Vehicle (DMC)	*p*-Value Compared to Doxorubicin (DMC)
Dedifferentiated liposarcoma	UZLX-STS124^DDLPS^	Vehicle	268.7	663.8	**0.0005**	261 (196–326)	**0.003**	n/a	>0.9999
		Doxorubicin	416.0	769.9	**0.002**	295 (211–379)		>0.9999	n/a
		Plocabulin	661.6	726.8	0.1641	140 (83–197)		**0.0186**	**0.0042**
	UZLX-STS112^DDLPS^	Vehicle	395.6	897.0	**0.0313**	214 (182–246)	**0.0002**	n/a	n/a **
		Doxorubicin	580.7	904.7	n/a **	221 (n/a **)		n/a **	n/a
		Plocabulin	375.0	431.8	0.2031	110 (82–138)		**0.0043**	n/a
Leiomyosarcoma	UZLX-STS22_2F^LMS^	Vehicle	236.9	660.2	**0.001**	287 (252–322)	**<0.0001**	n/a	0.674
		Doxorubicin	327.0	662.2	**0.0156**	218 (146–290)		0.674	n/a
		Plocabulin	337.5	87.1	**0.001**	27 (19–35)		**<0.0001**	**0.0091**
	UZLX-STS111^LMS^	Vehicle	501.2	958.8	**0.0039**	197 (163–231)	**0.0003**	n/a	n/a **
		Doxorubicin	552	1249.4	0.125	199 (142–256)		n/a **	n/a
		Plocabulin	483.8	395.9	**0.0039**	83 (77–89)		**0.001**	**0.0068**
Undifferentiated sarcoma	UZLX-STS84X^US^	Vehicle	249.2	570.6	**0.0078**	238 (190–332)	**0.0004**	n/a	>0.9999
		Doxorubicin	222.4	489.4	**0.0156**	261 (84–597)		>0.9999	n/a
		Plocabulin	259.6	217.1	**0.0371**	90 (65–155)		**0.0013**	**0.0041**
Intimal sarcoma	UZLX-STS84X^IS^	Vehicle	373.8	967.0	**0.0002**	265 (214–316)	n/a	n/a	n/a **
		Doxorubicin	316.8	n/a **	n/a **	n/a ** (n/a **)		n/a **	n/a
		Plocabulin	311.5	375.6	0.1099	130 (102–158)		**<0.0001** ***	n/a
*CIC*-rearranged sarcoma	UZLX-STS134^CIC^	Vehicle	894.3	1994.9	n/a **	363 (289–438) *	**0.0167**	n/a	>0.9999
		Doxorubicin	1594.5	574.1	n/a **	227 (n/a **)		>0.9999	n/a
		Plocabulin	1463.4	470.0	**0.0156**	40 (24–58) *		**0.0704**	0.6496

CI: confidence interval; DMC: Dunn’s multiple comparisons test; n/a: not available; WMP: Wilcoxon matched-pairs signed-rank test. * range instead of 95% confidence interval; ** not available due to insufficient tumours remaining on day 22 for analysis; *** Mann–Whitney U test instead of Dunn’s multiple comparisons tests. Significant *p*-values are highlighted in bold.

## Data Availability

The data used to support the findings of this study are available from the corresponding author upon request.

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
