# Peer review of "Plocabulin, a Novel Tubulin Inhibitor, Has Potent Antitumour Activity in Patient-Derived Xenograft Models of Soft Tissue Sarcoma"

_ijms, 2022, doi:10.3390/ijms23137454_

Round 1

Reviewer 1 Report

The authors described the role of plocabulin, using patient-derived xenograft models of soft tissue sarcoma. Please see my comments.

1) I think it should be interesting if the authors compare the antitumor effect between plocabulin and eribulin because eribulin is also a tubulin inhibitor.

2) Fig.1 I wonder if the characteristics of xenograft model tumours are similar to patients' original tumours, especially for 124DDLPS, LMS.

124DDLPS: MDM2 is negative in the mouse model while positive in the patient.

LMS: SMA and caldesmon are positive in cytoplasm in the mouse, while nucleus in the patient.

3) I don't understand the difference of IHC findings between the three groups. Especially, the pictures in plocabulin may be replaced because there are few cells in the pictures.

4) I wonder if the index of ki67 was not a difference between the three groups. Especially, in the LMS and CIC sarcoma model, the tumour volume was significantly decreased. 

5) Did you confirm the appropriate dose of doxorubicin? Why did you decide on 3mg/kg?

Reviewer 2 Report

In general, it is not easy to follow and understand what the authors did and why. The authors should try to improve the paper to be clear for all the readers. I understand that this paper is the continuation of another, but the authors must report all the necessary information so that the reader is able to follow the logical thread of what has been done and why.

Specific comments:

·       Lines 41-43: it seems that chemotherapy is not used for these tumours.

·       Line 60: references should be provided.

·       Line 64: could the authors specify which tumours cells?

·       Line 66-68: could the authors be more precise? Which tumour?

·       Line 72: which solid tumours?

·       Line 76: could the authors report these 7 models?

·       Line 79: the authors should list these 7 PDX models.

·       Lines 81-82: here, the authors reported that these models were published in reference 22. I suggest to clearly report what was published in this previous paper in the introduction.

·       Lines 82-83: why did the authors use PDX models at different passages?

·       Figure 1: The authors should present all the data. It is questionable not to have MDM2 IHC of DDLP tumour.

·       Figure 1: UZLX--STS124DDLPS: It seems to me that MDM2 positivity was lost in p.11.

·       Figure 1: In LMS models (p.20 and p12), the positivity is clearly not comparable to that of the original tumour. 

·       Line 88: could the authors explain what are these 41 treated mice? how many mice were allocated in each group? How many mice for each of the 7 models?  I suggest adding a scheme.

·       Section 2.4: could the authors quantify necrotic/degenerative area and compare it among the different conditions for all the models?  

·       Figure 4: I noticed that several data are missing.

·       Line 180: the authors reported “Antitumour activity of the experimental agent was observed in all models and all histological subtypes.” This part should be better explained. The authors demonstrated only a decrease of the tumour volume in all the models. However, it should be considered the different number of mice used etc.

·       Line 256: The authors should improve the limitation paragraph. For example, the lack of different data may have affected the interpretation of the results.  

·       Section 4.2: further information about the models should be provided. The reader should have enough information to understand what the authors have done.

·       Lines 284-285: “UZLX-STS112DDLPS, -STS124DDLPS, -STS22FLMS, -STS111LMS, -284 STS84XUSTS, -STS122FIS and -STS134CIC” are not clear.

·       Line 288: “least two passages” is not clear.

·       Lines 306-308: the authors reported that they implanted bilaterally and subcutaneously with tumour fragments 140 mice, generating 211 tumours. Why 211 tumours?

·       How many patients were enrolled? Inclusion/exclusion criteria are missing. Did the authors enrol 140 patients?

·       Lines 309-311: why did the authors chose these concentrations?

·       Lines 309-311: how many mice were allocated in each group? How many mice for each of the 7 models?  

·       Line 313: Why 22 days?

·       Section 4.5: dilution of the antibodies should be added.

·       Lines 392-393: “after having demonstrated the potential utility of plocabulin in gastrointestinal stromal tumour”. This part should be deleted as this manuscript is not focused on gastrointestinal stromal tumour.

·       Lines 396-397: the authors only studied plocabulin. Thus, the conclusions should be focused only on plocabulin. The authors did not evaluate other tubulin inhibitors etc. 

Round 2

Reviewer 2 Report

No additional comments